# Assessment of Mechanical, Chemical, and Biological Properties of Ti-Nb-Zr Alloy for Medical Applications

**DOI:** 10.3390/ma14010126

**Published:** 2020-12-30

**Authors:** Viktoria Hoppe, Patrycja Szymczyk-Ziółkowska, Małgorzata Rusińska, Bogdan Dybała, Dominik Poradowski, Maciej Janeczek

**Affiliations:** 1Centre for Advanced Manufacturing Technologies—Fraunhofer Project Center, Department of Laser Technologies, Automation and Production Management, Faculty of Mechanical Engineering, Wrocław University of Science and Technology, 50-371 Wrocław, Poland; patrycja.e.szymczyk@pwr.edu.pl (P.S.-Z.); malgorzata.rusinska@pwr.edu.pl (M.R.); bogdan.dybala@pwr.edu.pl (B.D.); 2Department of Animal Physiology and Biostructure, Division of Anatomy, Faculty of Veterinary Medicine, Wrocław University of Environmental and Life Sciences, 50-375 Wrocław, Poland; dominik.poradowski@upwr.edu.pl (D.P.); maciej.janeczek@upwr.edu.pl (M.J.)

**Keywords:** Ti-13Nb-13Zr, mechanical properties, XPS, corrosion resistance, cytotoxicity

## Abstract

The purpose of this work is to obtain comprehensive reference data of the Ti-13Nb-13Zr alloy base material: its microstructure, mechanical, and physicochemical properties. In order to obtain extensive information on the tested materials, a number of examination methods were used, including SEM, XRD, and XPS to determine the phases occurring in the material, while mechanical properties were verified with static tensile, compression, and bending tests. Moreover, the alloy’s corrosion resistance in Ringer’s solution and the cytotoxicity were investigated using the MTT test. Studies have shown that this alloy has the structure α’, α, and β phases, indicating that parts of the β phase transformed to α’, which was confirmed by mechanical properties and the shape of fractures. Due to the good mechanical properties (E = 84.1 GPa), high corrosion resistance, as well as the lack of cytotoxicity on MC3T3 and NHDF cells, this alloy meets the requirements for medical implant materials. Ti-13Nb-13Zr alloy can be successfully used in implants, including bone tissue engineering products and dental applications.

## 1. Introduction

The selection of the orthopaedic material is critical for its function. Material for implants replacing hard tissues should provide excellent mechanical properties, especially the Young’s modulus and compressive strength. At the same time, it should be biocompatible, without any cytotoxic effect. The biggest group of materials used for orthopaedic implants are metals: commercially pure titanium, alloys based on titanium or cobalt-chromium, and stainless steel [1]. In long-term research and practice, titanium and its alloys have proven to adapt well to a bone, especially titanium and Ti-6Al-4V alloy, thus becoming the most common in implants [2,3,4]. Although very popular, these so-called first-generation alloys are not ideal and bring unwanted effects during the post-implantation period. The biggest problem is the stress-shielding effect caused by the high Young’s modulus around 110 GPa when compared with the modulus of a natural bone varying between 15 and 30 GPa [5,6]. Moreover, the alloying elements (Al, Fe, and V) have been reported, to some extent, to cause toxic and allergic reactions [7,8].

New material development research processes are constantly searching for new and better materials for implant manufacturing. The biggest challenge in developing metallic implants for bones is to find a material with a modulus of elasticity similar to that of the neighbouring bone tissue and at the same time biocompatible in long-term exposition.

New trends in the developing titanium alloys for applications in biomedical engineering lead to alloys consisting of nontoxic and non-allergic alloying elements while maintaining excellent mechanical properties (particularly Young’s modulus to strength ratio) [9,10], [11]. The elastic modulus of recently developed alloys ranges from 55 to 85 GPa, which is much less than for commonly used alloys [12]. These so-called second-generation titanium alloys have higher beta phase content in the microstructure and more biocompatible alloying elements. The alloying elements from the first-generation alloys are replaced by others: V is replaced by Nb, Fe, and Mo, while Al is replaced by Ta, Hf, and Zr. New β-titanium alloys with Nb, Ta, and Zr as the main alloying elements constitute an emerging group of alloys for biomedical applications as they offer lower values of Young’s modulus (≈60 GPa), which is now more comparable to the mechanical properties of bones and higher biocompatibility than generally used Ti-6Al-4V and CP Ti [13]. It is assumed that elements such as Ta, Nb, Zr, and Hf may have favourable bio-neutrality generally attributed to titanium [14,15].

Literature reports indicate that β-titanium alloys have better strength and lower modulus of elasticity than α or α + β Ti alloys. For this reason, in recent years, more and more attention has been devoted to the development of a new generation of β Ti alloys consisting only of biocompatible components [16,17,18]. The Ti-Nb-Zr system was developed based on excellent properties of the two-component Ti-Zr system introduced in the early 1990s [11,19,20]. The undoubted advantage of alloys based on the Ti-Nb-Zr system is their high corrosion resistance because the passive layers forming these elements are permanent, their adhesion to the metallic substrate is good, and the tissue dissolution rate in the surrounding physiological fluids is low [11,17,21]. The Ti-13Nb-13Zr alloy is passive in simulated physiological conditions, such as Ringer’s [22,23,24] and Hank’s [25] solutions, artificial saliva solution and phosphate-buffered saline (PBS) [26]. However, it shows activity in hydrochloric acid solutions [27]. Another research indicates that alloys based on the Ti-Nb-Zr system are potentially more resistant to environmental factors in orthopaedic applications than dental [28].

From the practical side, the main difference between the first and second generation of titanium alloys is that the clinical community has much more experience and long-term evaluation data showing titanium and its traditional alloys as a safe and functional material for medical applications, and therefore, it is more often applied. In addition, R&D costs and expenses related to regulatory approval of the second-generation titanium alloys and its modifications are assumed to be more favourable than for the group consisting of Ta, Nb, Zr, and Hf, affecting the industry development and the clinical introduction of the new implant materials [29]. Nevertheless, the research for better orthopaedic implant materials continues and our input—the assessment of mechanical, chemical, and biological properties of Ti-Nb-Zr alloy—is presented below.

## 2. Materials and Methods

For the experiments, samples of the Ti-13Nb-13Zr ASTM F-1713:2008 alloy (Shaanxi Yuzhong Industry Development Co., Baoji, China) were delivered in the form of a cylinder Ø = 40 mm and L = 200 mm. This material was used to prepare standardised samples for mechanical testing in order to determine tensile, compressive, and flexural strength. To obtain the desired geometries, the Wire Electrical Discharge Machining method was used. Then, prefabricated rods and plates with reduced size were prepared and adjusted to the required geometries using a CNC lathe ST-10 (Haas Automation Inc., Oxnard, CA, USA) and CNC milling machine VF-1 (Haas Automation Inc.). For full characterisation of the investigated alloy—to confirm possibilities of its application as an implantation material—the microstructure, electrochemical, and biological properties were investigated using samples in the form of pellets with dimensions Ø14 mm × 2 mm.

### 2.1. Material Properties Analyses

Macroscopic images were taken using a VHX digital microscope (Keyence, Osaka, Japan). The microstructure was investigated with a Scanning Electron Microscope EVO MA 25 (Zeiss, Oberkochen, Germany). The samples in the form of plates were prepared for SEM analysis by grinding on abrasive from #200 to #2000 and polishing using colloidal SiO_2_ to obtain a mirror surface and then etched with Kroll reagent (92 mL distilled water + 6 mL nitric acid + 2 mL hydrofluoric acid). Phase analysis was performed using an X-ray powder diffractometer MiniFlex 600 (Rigaku, Tokyo, Japan) and Smart Lab Studio II software (Rigaku, Tokyo, Japan). XRD was conducted using CuKα radiation and a scanning speed of 2°/min from 30° to 90°. Square surfaces (20 mm × 20 mm) were polished using #2000 abrasive and colloidal SiO_2_ with polishing cloth to ensure that the diffractometry was performed in the interior structure.

### 2.2. X-ray Photoelectron Spectroscopy

X-ray photoelectron spectroscopy (XPS) spectra were obtained with a Kratos Axis Supra spectrometer (Kratos-Shimadzu, Tokyo, Japan) with a monochromatic Al Kα radiation source (1486.7 eV). The instrument was calibrated for BE = 84.0 eV ± 0.1 eV for the 4f7/2 line of metallic gold. The spectrometer dispersion was adjusted to give BE = 932.62 eV for the Cu 2p3/2 line of metallic copper. The energy resolution was examined on a metallic silver sample. Survey(wide) spectra were gathered with a quality corresponding to the full width at half maximum (FWHM) parameter for Ag 3d line equal to 0.71 eV at energy step size 0.5 eV. For high-resolution spectra, the FWHM parameter for the Ag 3d line was equal to 0.58 eV at a step size of 0.1 eV. The peaks fitting was performed using CasaXPS software version 2.3.18 (Casa, Osaka, Japan) on a Shirley background.

### 2.3. Mechanical Properties Analyses

The strength tests to determine the mechanical properties of the Ti-13Nb-13Zr alloy were executed on the universal Instron 3385 (Instron, Norwood, MA, USA) testing machine with a maximum load of 150 kN. Detailed data on standards, sample dimensions, and mechanical test parameters are presented in Table 1. A static tensile test was performed based on the ASTM E8/E8M-16a standard [30]. The test was performed for traverse constant speed of 2 mm/min using a video extensometer. The end of the test occurred when the sample was broken. The test was carried out for 10 samples. A static compression test was carried out based on ASTM E9-09 standard [31]. The test was performed for a traverse constant speed of 1 mm/min. The end of the test occurred when the sample was broken. The test was carried out for 10 samples. A static bending test was carried out based on the ASTM E290-14 standard [32]. The distance between the lower rollers was chosen according to the standard, and it was set at 19 mm. Samples were bent at a constant speed of 1 mm/min. The criterion for the end of the test was a load decrease of about 10% after the sample break.

### 2.4. Electrochemical Properties Analyses

Electrochemical measurements were executed on specimens with the working area of 0.75 cm^2^, in a Ringer’s solution (8.6 g/dm^3^ NaCl, 0.3 g/dm^3^ KCl, 0.243 g/dm^3^ CaCl_2_), at room temperature (25 °C), using an Atlas Sollich 1131 electrochemical potentiostat (Atlas Sollich, Rębiechowo, Poland). The working electrode was grinded with SiC #2000 abrasive paper before each experiment. Corrosion resistance measurements were executed by potentiodynamic polarisation using a platinum electrode as a counter electrode (CE), calomel electrode as a reference (SCE) and sample as a working electrode (WE). Before each experiment, the Open Circuit Potential (OCP) was recorded after 120 min of resting in the electrolyte when the potential changes were under 1 mV/min. Tafel curves were recorded by changing the electrode potential from −0.3 to +1.5 V versus the OCP potential with a rate of 1 mV/s. The Stern–Geary Equation (1) was used to calculate the polarisation resistance:(1)$$R_{p}=\;\frac{\beta_{a}\;\times \;\beta_{c}}{2.303\;\times \;(\beta_{a}\;+\;\beta_{c})\;\times \;i_{corr}}$$
where: *β_a_* is the anodic Tafel slope, *β_c_* is the cathodic Tafel slope, and *i_corr_* is the corrosion current density. Tafel curves were recorded for 5 measurements, the results were averaged, and the deviation was reported.

### 2.5. Biological Properties Analyses

Cytotoxicity studies were performed according to the MTT protocol based on EN ISO 10993-5:2012 [33] and EN ISO 10993-5:2009 standard [34]. The research was carried out on the normal human dermal fibroplast (NHDF) cell line (American Type Culture Collection, Manassas, VA, USA) and the MC3T3-E1 mouse preosteoblast line (American Type Culture Collection). Cells of the tested lines were applied to Eppendorf 96-well culture plates. Then, 100 µL of cells suspended in medium at a density of 1 × 10^5^/mL (corresponding to 1 × 10^4^/well) were added to each well. Extracts from the tested samples were prepared in the proportion: 0.2 g of sample per 1 cm^3^ of culture medium DMEM (Dulbecco’s Modified Eagle Medium) supplemented with fetal bovine serum (FBS) for 24 h in a controlled environment (37 °C, 5% CO_2_, full humidity). The concentrations tested were 100%, 50%, 25%, and 12.5% of the initial concentration of the extract. In this study, two controls were used: positive (mitomycin C at concentration of 2 µg/mL) and negative (untreated cells). The blank was a culture medium with the composition as mentioned above. Then, the obtained extracts were introduced into the cell culture described above and incubated for 72 h in 5% CO_2_ at 37 °C. For better clarity of results, 10 independent replicates, for four concentrations of the test sample extract in the medium, were conducted. After the incubation, 100 µL of isopropyl alcohol was added to the wells to dissolve the violet formazan crystals, which appear in metabolically active cells. The absorbance of the formazan solution, after gentle mixing, was read at wavelength λ = 570 nm and reference wavelength λ = 720 nm in Multiscan Go spectrophotometer (ThermoFisher, Waltham, MA, USA). The percentage of live cells was calculated by comparing the absorbance value of the test samples with the value of the control (untreated cells). Control absorbance was assumed as 100%.

## 3. Results

### 3.1. Material Properties Analyses

A typical annealed Ti-13Nb-13Zr SEM microstructure is shown in Figure 1. It contains a finely dispersed acicular α martensite in β matrix; very soft prior β grain boundaries are also visible. In morphological terms, the α-phase is enriched in Zr (neutral element), and the β-matrix is enriched in Nb (β-stabilised element). The microscopic image indicates that the structure was not aged.

The presence of two phases was verified through X-ray diffraction (XRD) investigations. Figure 2 summarises typical X-ray diffraction pattern for Ti-13Nb-13Zr. All the major peaks were indexed as α-Ti and β-Ti. Bragg’s peak indexation denotes the presence of α-Ti (PDF ICDD 00-044-1294) and β-Ti (PDF ICDD 01-089-3726) phases.

The chemical composition specified by the Ti-13Nb-13Zr manufacturer is presented in Table 2. These results are compared with the requirements of the ASTM F1713 standard [35].

It was found that the chemical composition meets these requirements, and the contents of individual elements are within the ranges required.

### 3.2. X-ray Photoelectron Spectroscopy

A widescan XPS survey of Ti-13Nb-13Zr titanium alloy showed major peaks domination at Ti, O, and C, additionally Zr and Nb, and moreover, there are detected peaks from contaminants such as N and F [36]. In the high-energy part of the XPS spectra, there are wide MNN Auger bands of Zr and Nb (≈1105 eV and ≈1085 eV, respectively), LMM Ti Auger bands (≈1065 eV), KVV Auger bands of C, N, O and F (≈990 eV, ≈1105 eV, ≈990 eV, ≈858 eV respectively) (Figure 3). Moreover, on the widescan, a dominated doublet peak is visible at ≈460 and ≈465 eV for Ti, which can be assigned to Ti oxide [37,38]. The presence of small amounts of nitrides and carbides is an effect of machining of titanium surfaces in the presence of air and organic lubricants [39].

Relative atom concentrations and binding energies of all elements are summarised in Table 3.

Deconvoluted XPS data revealed the presence of Ti, Nb, and Zr but also O, as expected, as that element is commonly adsorbed on Ti surfaces. The spectra are dominated by Ti and O due to the naturally formed TiO_2_ layer. Deconvolution of the O spectra (Figure 4) showed the presence of hydroxides, which frequently shows spectral components for 531 eV, which may be assigned to OH. The Ti 2p spectrum obtained from the Ti gives 4 doublets, corresponding to the valences Ti^0^ (metallic state), Ti^2+^ (TiO), Ti^3+^ (Ti_2_O_3_), and Ti^4+^ (TiO_2_) [40], as shown in Figure 5.

Deconvolution of spectra of niobium showed the presence of peaks that corresponds to valences Nb^0^ (metallic state), Nb^4+^, and Nb^5+^ (Nb_2_O_5_) [41] (Figure 6). Zr 3d spectra obtained from Zr showed peaks that correspond to the valences Zr^0^ (metallic state) and Zr^4+^ (Zr_2_O) [42] (Figure 7).

### 3.3. Mechanical Properties Analyses

The results obtained for the mechanical properties present curves with a typical shape, which is characteristic for materials with proof stress (stress is taken at which 0.2% plastic deformation occurs). During static tensile strength, the parameters included in the ASTM F1713-08 standard [35] were determined. The average UTS (Ultimate Tensile Strength) for the tested Ti-13Nb-13Zr alloy samples was 733.9 ± 7.5 MPa, YS 0.2% (Yield Strength, proportional elastic limit with an elongation of 0.2%) was 555.6 ± 17.3 MPa, and A (Elongation) was 19.6 ± 1.6%. One of the most important parameters from the point of view of biomedical engineering was also determined—the value of Young’s modulus was 84.1 ± 1.8 GPa. Moreover, during the static compression test, parameters such as CS (compressive strength) and YSc (compression yield strength) were determined. The following results were obtained: CS was 1378.9 ± 55.8 MPa and YSc was 757.3 ± 34.9 MPa, respectively. During the three-point bending test, the FS (flexural strength) was determined and was 1708.2 ± 43.9 MPa. Table 4 presents gathered results compared with ASTM F1713 and supplier certificate.

The obtained results are similar to those provided by the manufacturer. The deviations may result from the research method used. Additionally, the supplier’s certificate did not provide values that correspond to the results obtained during static compression and bending tests. It is worth mentioning that the difference concerning the requirements of the ASTM F1713 standard and results from this research appeared due to the treatment (annealing) applied by the supplier other than that described in the standard.

The stress–strain curves of the investigated alloy from tensile and compression tests are presented in Figure 8 and Figure 9, respectively. When analysing these figures, it can be seen that the curves take the typical shape that is characteristic of materials that do not exhibit a clear yield point. The obtained curves for all static tensile test samples were similar to the linear characteristics—the curves were within the range of the Hooke’s law, which made determining Young’s modulus possible.

Results of three-point bending and curves obtained in this test are found on the flexural strength versus deflection curve (Figure 10). The figure below shows the area of elastic deformation up to 1200 MPa, plastic deformation 1600–1800 MPa, and cracks marked by a clear decrease in strength.

Samples that were subjected to mechanical properties tests were analysed using fractography methods. Samples after static tensile test were characterised by distributive beam fracture. It is formed by detaching the upper and lower parts of the sample from each other, and its orientation is perpendicular to the direction of the formation of the largest tensile stress. Due to the fact that before the sample breaks, plastic deformation also occurs, this fracture is not completely brittle, and a neck is formed during stretching. A brittle fracture can be observed in the central part of the sample (Figure 11). It is characterised by a shiny, irregular surface. Around the brittle fracture, a slip fracture may be seen with a matte, smooth surface.

The fracture resulting from the static compression test is classified as a slip fracture due to the location of the crack at an angle of 45% for normal forces. This means that during compression, the sample was subjected to plastic deformation. It can be seen that the sides are deformed in characteristic barrel shape and cracked diagonally during destruction. Figure 12 presents the cross-section, created by the fracture of the sample, taken transversely to the sample axis along with an oblique photo of this cross-section with a visible slip direction.

After three-point bending, it was observed that samples have a characteristic brittle fracture, i.e., they cracked without visible deformation. The characteristic appearance for this type of fracture is also visible—grainy and shiny (Figure 13). A crack appeared in the center of the sample between the lower loading rollers.

### 3.4. Electrochemical Properties Analyses

Potentiodynamic polarisation was used for corrosion resistance measuring. The parameters of the corrosion process from Tafel curves are presented in Table 5. An example curve is presented in Figure 14.

As can be seen, the polarisation curve of the investigated alloy shows a broad passive area from 0 to 1.5 V when the current anodic density remained unchanged as the potential increased. This area indicates protective oxide film formation to prevent the continual metal from dissolving and thus inhibiting anodic process. For these samples, the anodic reaction was limited to the dynamic of the corrosion degradation—the mechanism of anodic protection. The results indicate that the passive film consists of a single layer and its resistance, R_p_, was equivalent to 523.97 ± 170.97 kΩ cm^2^.

### 3.5. Biological Properties Analyses

Positive results were achieved in the verification of the biological properties. Taking into account the cellular behaviour of MC3T3 and NHDF cells on the Ti-13Nb-13Zr alloy, both cell lines show a high survival rate and lack of morphological changes: no intra-plasmatic granules, no cell lysis, and culture density is comparable to a negative culture. Detailed data from the MTT test for MC3T3 and NHDF are included in Table 6 and Table 7, respectively.

For MC3T3 osteoblast cells, depending on the concentration of the extract, the cell survival rate is from 111.67 ± 2.44% for 100% extract to 128.32 ± 4.22% for 12.5% extract. A similar case can be observed for NHDF cells and their cell survival rate is from 111.75 ± 3.04% for 100% extract to 129.05 ± 9.08% for 12.5% extract. Tested Ti-13Nb-13Zr alloy has not shown any cytotoxicity to MC3T3 and NHDF cell lines.

## 4. Discussion

Beta titanium alloys are promising materials for medical implants due to their alloying β-stabilising elements, which are also biocompatible and nontoxic. They exhibit a lower Young’s modulus and better corrosion resistance than α and α + β alloys. The Ti-13Nb-13Zr alloy is a highly appreciated near β-type titanium alloy for biomedical applications since its development by Davidson et al. in the 1990s [43]. Titanium alloys exist in several crystallographic forms. At room temperature, titanium has an HCP (hexagonal close-packed) crystal structure called the α phase. This structure has the ability to transform at 883 °C to a BCC (body-centered cubic) crystal structure, referred to as the β phase. In the case of the Ti-13Nb-13Zr alloy, its β-stabiliser is the niobium, which affects the microstructure and phase composition resulting in stability of the β phase at a lower temperature [44]. Microscopic images obtained by scanning electron microscopy indicate the presence of the acicular phase α with martensitic origin (transformed β) in the β phase matrix (Figure 1). Aijt et al. [17] and Khorasani et al. [45] indicate that the microstructure at room temperature contains α’, α, and β phases—implying that some of the β phase transformed to α’, while the rest of the β transformed into the equilibrium of α + β phase, which confirms the microstructure of the tested material in this research. It has been recognized by Liu et al. [10] that the metastable phase α′, formed martensitically by quenching, has the same crystal structure as the α phase. The α′ phase could dissolve the higher content of β stabilisers than the stable α phase, and the more β stabilisers, the lower the Young’s modulus [10]. This statement confirms our XRD results and the presence of the phases mentioned above.

As a result of the intended application of the studied titanium alloy as a material for bone implants, one of the key factors to be considered is the Young’s modulus, which should achieve values similar to bone parameters. Our studies show that E = 84.1 ± 1.8 GPa, which in turn is convergent with the results obtained by other scientists [16,17,37] in the same heat treatment condition of Ti-13Nb-13Zr alloy (Table 8). The value of the Young’s modulus of Ti-Nb-Zr alloys depends on the alloy’s microstructure (β phase has a lower Young’s modulus than the α phase), which in turn depends on the amount and type of alloying elements and the processing techniques. Thermal or mechanical processing causes changes in the microstructure of materials. In cold-worked materials, it results in deformation that makes the material stronger and harder, less ductile, and more chemically reactive. β-stabiliser elements (Nb, Zr, Hf, Ta, Mo, etc.) result in a lower elastic modulus [10], as the tested near-beta alloy, and another based on Ti-Zr or Ti-Nb-Zr, which exhibit the Young’s modulus [11,16,46,47] lower by up to 30% compared with α-alloys [14,48].

The characterisation of mechanical properties of the Ti-13Nb-13Zr alloy was performed with tensile, compression, and bending tests. Most scientists only carry out a tensile test to determine the mechanical properties, which does not show the alloy’s mechanical properties under different load conditions. Therefore, we decided to extend the scope of mechanical tests with a static compression test and three-point bending, due to the fact that the parameters determined in these tests will allow for implementing their values into experimental models—for example, for the finite element analysis. Mechanical properties of materials are crucial when designing load-bearing orthopaedic or dental implants. Implant failure is catastrophic because the fracture of any implant could be very dangerous for a patient. For example, during oral activities, implants are required to support axial loads around 120–200 N [49]. Observed fracture shows a mixed character. The area with features characteristic for the tough fracture is small. The occurrence of quasi-brittle fracture should be connected with the hexagonal α′ phase. It is assumed that increase of the ductility is caused by the transformation of the β phase to α’ martensite [16].

Many scientists point to the superior corrosion properties of the Ti-13Nb-13Zr alloy [26,50]. The corrosion resistance of passive metals depends on the chemical composition of the oxide layer. This is mainly due to the solubility of oxides formed on the top layer—Nb and Zr oxides have lower solubility than Al and V oxides [51]. The XPS analysis [52] showed that the passive layer of Ti-13Nb-13Zr alloy was formed by a mixture of TiO_2_, ZrO_2_, and Nb_2_O_5_ oxides. Titanium oxide is well known as a typical nontoxic, highly photochemically active semiconductor, which provides self-disinfecting properties for biomaterial surfaces [53]. The confirmation of the presence of oxides of the alloy elements on the surface of the material tested in this study clearly indicates that this material is excellent for use in medical implants due to its high biocompatibility properties and the possibility of molecules adhering through the presence of OH groups [33,54]. In the case of corrosion, tests carried out on the Ti-13Nb-13Zr in Ringer’s solution (pH 7.3–7.4) by other scientists [27] showed that corrosion current density at a temperature of 37 °C was 0.12 µA cm^−2^. In the same temperature, [22] showed that I_corr_ is 1.77 µA cm^−2^. These values are very close to the data obtained in this study: I_corr_ = 0.091 ± 0.033 µA cm^−2^. Slight differences may be due to the different temperatures at which the experiments were conducted, but the results are nevertheless very consistent. The resistance of the passivating film formed on Ti-13Nb-13Zr alloys is higher than in case of CP Ti or Ti-6Al-4V alloy, which is indicative of the higher corrosion [23]. This resistance is attributed to the passive TiO_2_ layer, which inhibits the absorption of corrosive ions [55]. The passive layer alters the surface potential and limits the number of charge carriers in the metal–electrolyte interlayer, protecting the substrate from corrosion damage. All mentioned sources [14,22,27] indicate an occurrence of long passive potential, highlighting superior corrosion resistance. The results achieved in our MTT tests are coherent with similar tests reported in the literature, which also prove the absence of cytotoxic effects of titanium, as opposed to most other metallic elements [15]. These data confirm that the samples do not present any cytotoxic effect. Previous studies have also confirmed the high cytocompatibility of Nb, Ta, Sn, and Zr as alloy elements [56], so the hypothesis is confirmed that the use of nontoxic elements results in the chemical composition of the alloy, which also has no cytotoxic properties. Further biological research using modified surfaces is necessary and will lead to a better understanding of biocompatibility of titanium alloys and other biological materials.

## 5. Conclusions

While designing and selecting biomaterials used in medical implants, it is crucial to make their mechanical, physical, chemical, and biological properties similar to those of living tissues. The comprehensive mechanical, physicochemical, and biological evaluation of Ti-13Nb-13Zr alloy was performed, and the results were compared with data from other scientists if those were available. The following main conclusions, supported by corresponsive data from other research, can be drawn:The microstructure at room temperature was confirmed by SEM and XRD investigations and was found to contain α, α’, and β phases, implying that some of the β phase transformed to α’.Mechanical properties obtained by the static tensile test are similar to the data found in the literature and conform to the requirements of ASTM F1713. For the annealed conditions: E = 84.1 ± 1.8 GPa, UTS = 733.9 ± 7.5 MPa, YS = 555.6 ± 17.3 MPa, and A = 19.6 ± 1.6%.XPS results confirmed the dominant presence of TiO_2_ on the examined surface and showed the presence of hydroxyl groups on the surface. Moreover, other oxides are present—Nb_2_O_5_ and Zr_2_O. The occurrence of oxides increases the biocompatibility of materials and hydroxide groups allow for covalent attachment of various molecules.As shown by our Tafel-plot extrapolations, the Ti-13Nb-13Zr in the Ringer’s solution spontaneously forms a passive layer, due to long passivation area on potentiodynamic polarisation curves.The toxic effect was assessed by a cytotoxicity MTT test, which proved that the Ti-13Nb-13Zr alloy did not present cytotoxic effect in the in vitro examination both for MC3T3 and NHDF cells.

The most important and totally new discovery presented by this research is the results obtained by the extended scope of mechanical tests with the static compression and the three-point bending. These values, hard to find in the literature, were determined and the following results were obtained: UCS = 1378.9 ± 55.8 MPa, YSc = 757.3 ± 34.9 MPa, and FS = 1708.2 ± 43.9 MPa. Together with these tests, another important factor came to light: the observed fractures that exhibit the occurrence of quasi-brittle fracture associated with the presence of the hexagonal α′ phase. The determined parameters allow for implementing their values into experimental models, for example, for the finite element analysis. These key values for potential biomedical applications constitute to be essential when designing load-bearing orthopaedic or dental implants.

## Figures and Tables

**Figure 1 materials-14-00126-f001:**
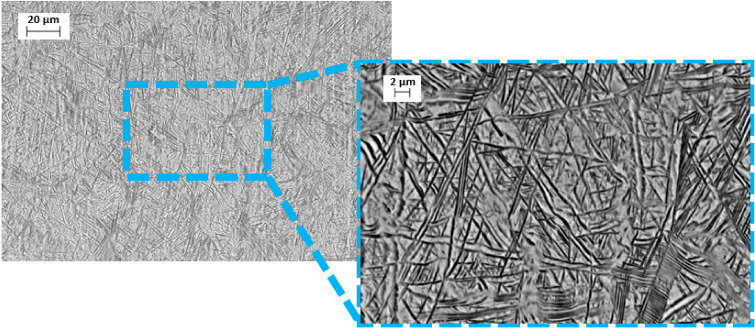
Microscopic image of Ti-13Nb-13Zr alloy microstructure, SEM, BSE.

**Figure 2 materials-14-00126-f002:**
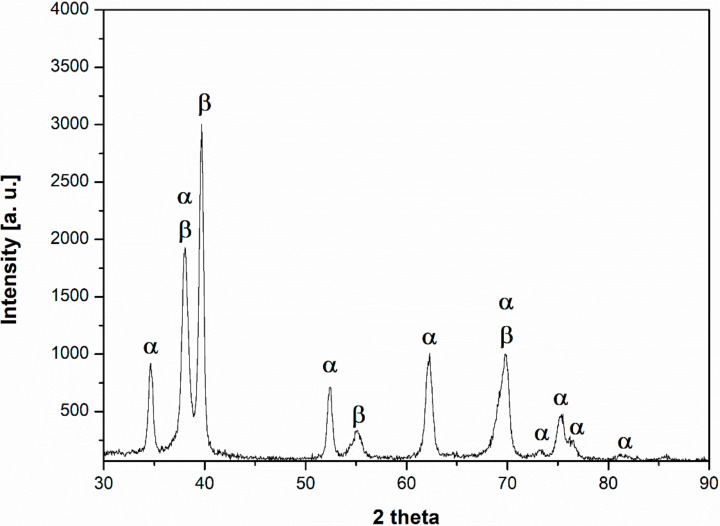
X-ray diffraction pattern with marked α and β phase reflections.

**Figure 3 materials-14-00126-f003:**
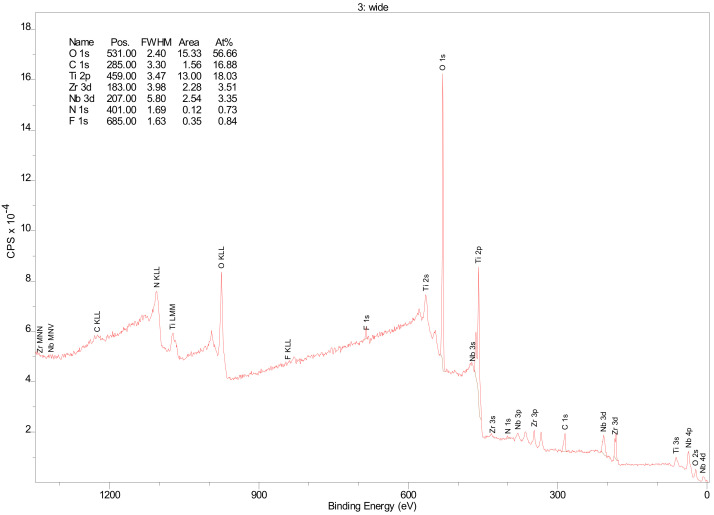
XPS widescan spectra of Ti-13Nb-13Zr titanium alloy sample after etching using Ar^+^ 3 min, 3 kV, 20 µA/cm^2^.

**Figure 4 materials-14-00126-f004:**
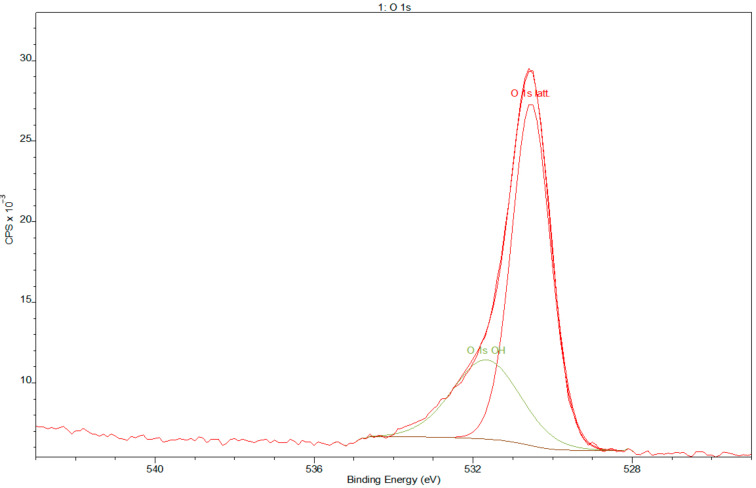
Deconvolution XPS spectra of oxygen.

**Figure 5 materials-14-00126-f005:**
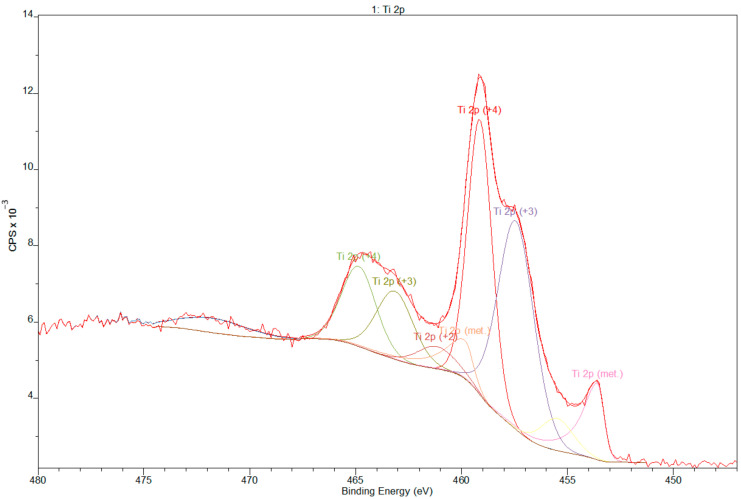
Deconvolution XPS spectra of titanium.

**Figure 6 materials-14-00126-f006:**
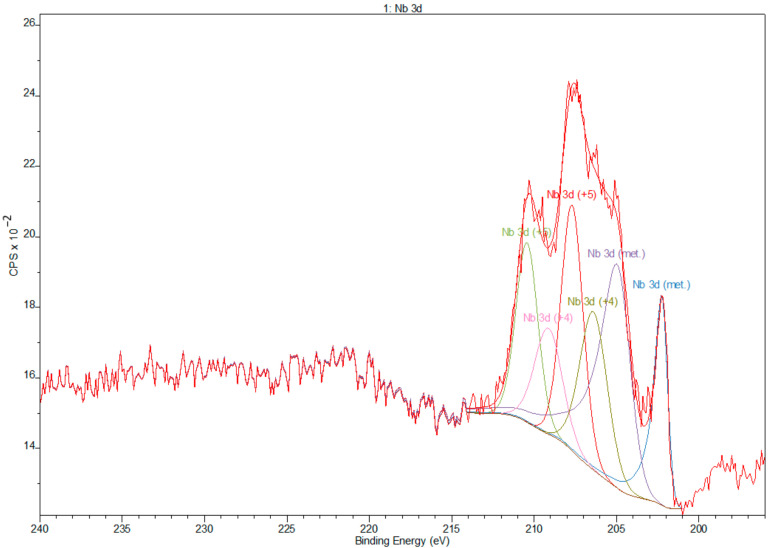
Deconvolution XPS spectra of niobium.

**Figure 7 materials-14-00126-f007:**
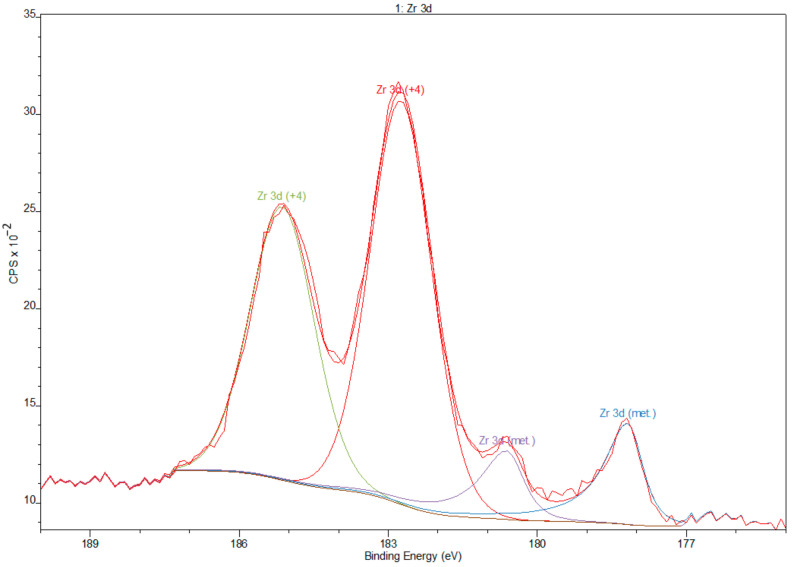
Deconvolution XPS spectra of zirconium.

**Figure 8 materials-14-00126-f008:**
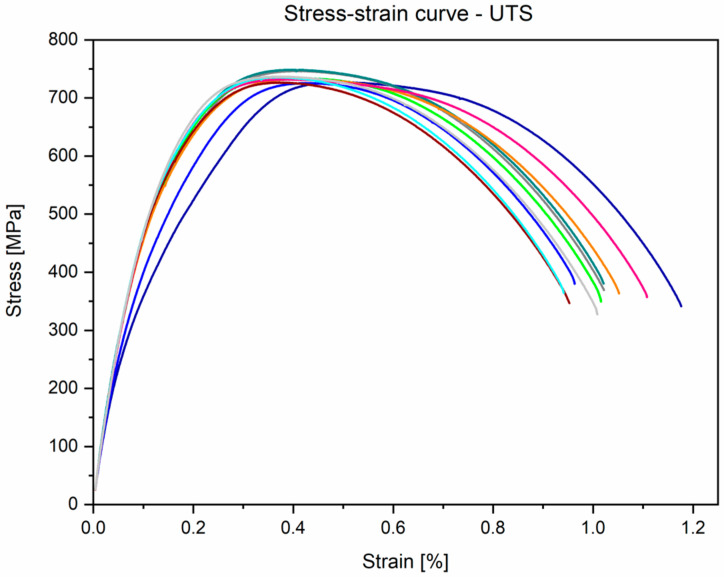
Experimental stress–strain curves obtained from the static tensile test for Ti-13Nb-13Zr alloy.

**Figure 9 materials-14-00126-f009:**
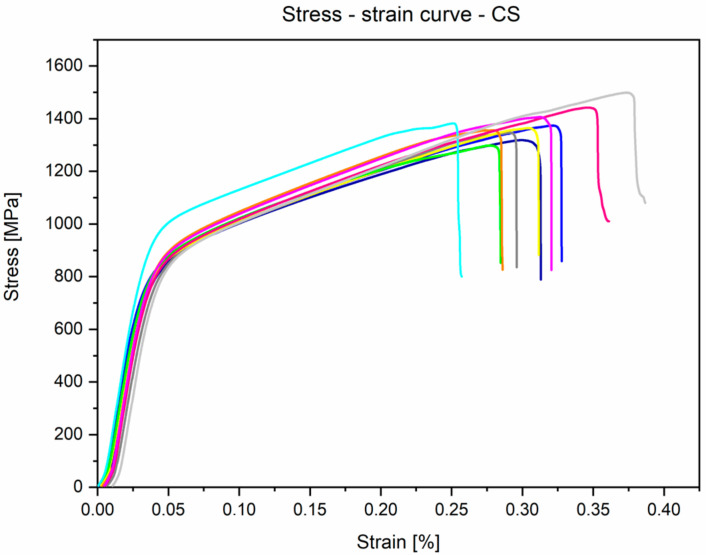
Experimental stress–strain curves obtained from the static compression test for Ti-13Nb-13Zr alloy samples.

**Figure 10 materials-14-00126-f010:**
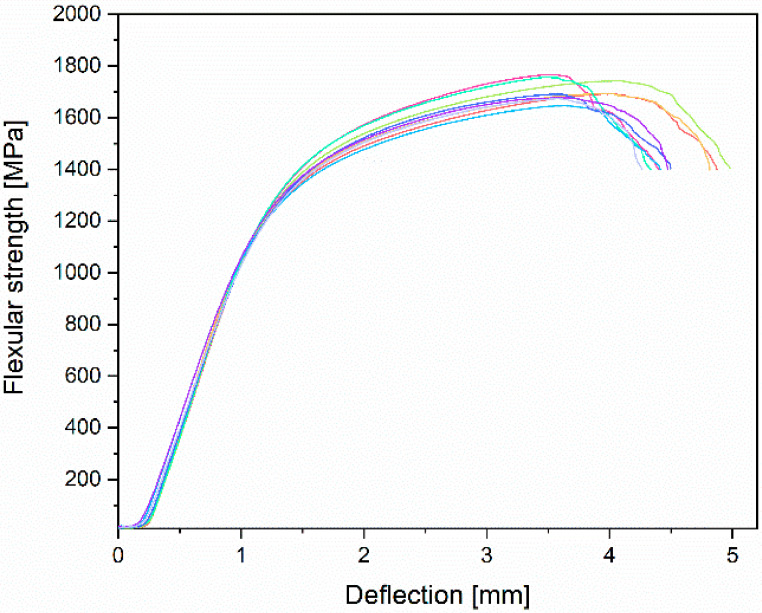
Experimental strength–deflection curves obtained from the three-point bending test for Ti-13Nb-13Zr alloy samples.

**Figure 11 materials-14-00126-f011:**
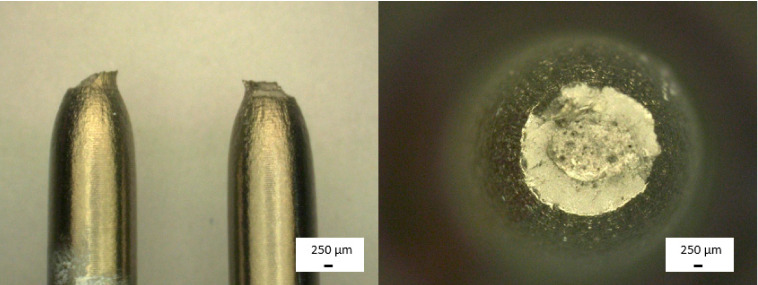
Light microscopy images of Ti-13Nb-13Zr samples after static tensile test.

**Figure 12 materials-14-00126-f012:**
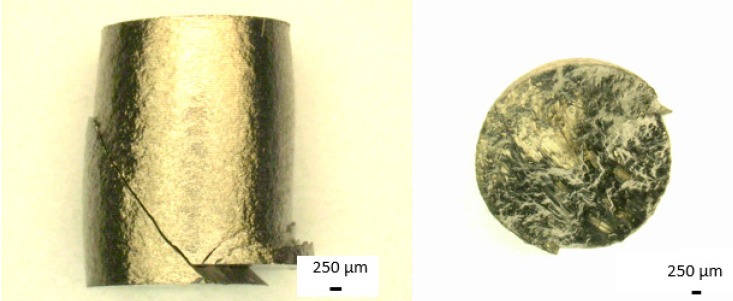
Light microscopy images of Ti-13Nb-13Zr samples after static compression test.

**Figure 13 materials-14-00126-f013:**
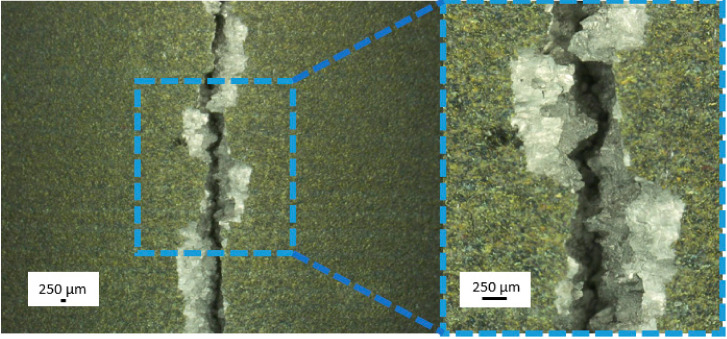
Light microscopy images of Ti-13Nb-13Zr samples after the three-point bending test.

**Figure 14 materials-14-00126-f014:**
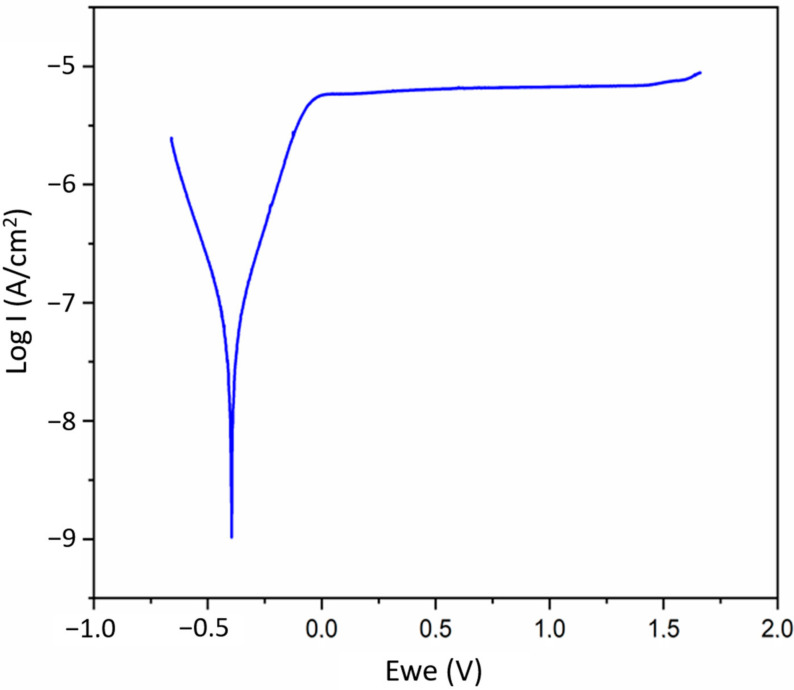
Potentiodynamic polarisation curve, recorded in Ringer’s solution at room temperature.

**Table 1 materials-14-00126-t001:** Parameters of mechanical tests of Ti13Nb13Zr alloy.

Type of Test	Measured Parameters	Standard	Dimensions	Traverse Speed	Number of Specimens
Static tensile test	UTS YS A E	ASTM E8/E8M-16a	G = 20.0 ± 0.1 mm D = 4.0 ± 0.1 mm R = 4 mm A_min_ = 24 mm	2 mm/min	10
Static compression test	CS YS_c_	ASTM E9-09	D = 6.5 ± 0.1 mm L = 12.5 ± 0.5 mm L/D = 2	1 mm/min	10
Static bending test	FS	ASTM E290-14	W = 20.0 ± 0.2 mm T = 3.0 ± 0.1 mm L = 75.0 ± 0.2 mm	1 mm/min	10

**Table 2 materials-14-00126-t002:** Chemical composition of Ti-13Nb-13Zr alloy in (wt%).

Elements	Ti	Nb	Zr	O (max)	Fe (max)	H (max)	C (max)	N (max)
ASTM F1713	77.46–74.46	12.50–14.00 ± 0.30	12.50–14.00 ± 0.40	0.15 ± 0.02	0.25 ± 0.10	0.012 ± 0.0020	0.08 ± 0.02	0.05 ± 0.02
Producer report	74.71	12.8	12.2	0.094	0.1	0.005	0.080	0.008

**Table 3 materials-14-00126-t003:** Binding energies of elements and their relative atom concentration from XPS widescan spectra.

Name	Position (eV)	FWHM	Raw Area	%At Conc
O 1s	531.00	2.40	318459	56.66
C 1s	285.00	3.30	26929.9	16.88
Ti 2p	459.00	3.47	256238	18.03
Zr 3d	183.00	3.98	36626.4	3.51
Nb 3d	207.00	5.80	41470.6	3.35
N 1s	401.00	1.69	2298.6	0.73

**Table 4 materials-14-00126-t004:** Mechanical properties of Ti-13Nb-13Zr alloy.

	UTS (MPa) (min)	YS (MPa) (min)	A (%) (min)	E (GPa)	CS (MPa)	YS_c_ (MPa)	FS (MPa)
ASTM F1713 (unannealed)	550	345	15	64–77	n.d.	n.d.	n.d.
Supplier Certificate (annealed *)	690	446	25	n.d.	n.d.	n.d.	n.d.
This research	733.9 ± 7.5	555.6 ± 17.3	19.6 ± 1.6	84.1 ± 1.8	1378.9 ± 55.8	757.3 ± 34.9	1708.2 ± 43.9

* 700 °C/1 h.

**Table 5 materials-14-00126-t005:** Polarisation parameters and corrosion rates evaluated with the Tafel method (E_corr_—rest potential, b_a_ and b_c_—anodic and cathodic Tafel slopes, I_corr_—corrosion current density, R_p_—polarisation resistance).

	E_corr_	b_a_	b_c_	I_corr_	R_p_
	mV	mV dec^−1^	mV dec^−1^	μA·cm^−2^	kΩ·cm^2^
mean	−366.83	208.33	191.64	0.091	523.97
SD	42.67	46.06	16.43	0.033	170.97

**Table 6 materials-14-00126-t006:** MTT test result for MC3T3 cells.

Test	Number of Samples	Concentration	Survival Rate (%)	Morphological Changes in Cell Cultures	Evaluation of Morphological Changes in Cultures	Cytotoxicity
Negative test	5	Untreated cells	97.33	No intra-plasmatic granules, no cell lysis was found.	0	none
Positive test	5	2 µg/mL	50.91	Cell culture almost or completely destroyed.	4	strong
Ti-13Nb-13Zr	5	100%	111.67 ± 2.44	No intra-plasmatic granules, no cell lysis was found. Culture density comparable to negative culture.	0	none
	5	50%	123.9 ± 4.65		0	none
	5	25%	125.39 ± 4.37		0	none
	5	12.5%	128.32 ± 4.22		0	none

**Table 7 materials-14-00126-t007:** MTT test result for NHDF cells.

Test	Number of Samples	Concentration	Survival Rate (%)	Morphological Changes in Cell Cultures	Evaluation of Morphological Changes in Cultures	Cytotoxicity
Negative test	5	Untreated cells	98.17	No intra-plasmatic granules, No cell lysis was found.	0	none
Positive test	5	2 µg/mL	51.67	Cell culture almost or completely destroyed.	4	strong
Ti-13Nb-13Zr	5	100%	111.75 ± 3.04	No intra-plasmatic granules, no cell lysis was found. Culture density comparable to negative culture.	0	none
	5	50%	116.5 ± 1.58		0	none
	5	25%	125.31 ± 4.65		0	none
	5	12.5%	129.05 ± 9.08		0	none

**Table 8 materials-14-00126-t008:** Comparison of the mechanical properties of Ti-13Nb-13Zr material with various heat treatment conditions.

	UTS (MPa)	YS (MPa)	A (%)	E (GPa)
This research (annealed)	733.9 ± 7.5	555.6 ± 17.3	19.6 ± 1.6	84.1 ± 1.8
Hardened [17,43]	798 ± 17	599 ± 24	20 ± 3	83±2
WQ [17,43]	703–786	433–554	21–29	64–77
WQ + Aged. [17,43]	994 ± 42	864 ± 43	13 ± 3	81 ± 4
WQ + DH [17,43]	1034 ± 5	906 ± 10	11 ± 1	83 ± 2
WQ + 50–75%CW [17,43]	1000–1055	900–1000	10–15	44–51
ST + WQ [16]	860	640	15	60
52% CW [16]	1280	1270	12	52

(WQ—water quenched; DH—diffusion hardened; CW—cold worked; ST—solution treated).

## Data Availability

The data presented in this study are available on request from the corresponding author.

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
