# Peer review of "Assessment of Mechanical, Chemical, and Biological Properties of Ti-Nb-Zr Alloy for Medical Applications"

_materials, 2020, doi:10.3390/ma14010126_

Round 1

Reviewer 1 Report

Scientific work on 18 pages contains 14 figures, 9 tables and 50 literary references.

     The article investigates the microstructure, electrochemical and biological properties of the commercial titanium alloy Ti-13Nb-13Zr (ASTM F-1713). The authors call this alloy a biocompatible titanium alloy of the second generation, which has a lower modulus of elasticity (≈ 84 GPa), which is closer to the values ​​of the elastic modulus of human bone tissue (15-30 GPa) compared to the modulus of elasticity of the previous generation of alloys (≈ 84 GPa).

  1. The content of the work is sufficiently consistent with the subject and profile of the journal.
  2. The originality and novelty of the material proposed by the authors of the article are at a satisfactory level, since a sufficiently studied commercial alloy is being investigated. Of interest is the discovery of quasi-brittle fracture associated with the presence of the hexagonal α 'phase.
  3. The theoretical and practical value of the work is due to the possibility of using the results obtained in the article in the conduct of research and development work related to the design of bearing orthopedic or dental implants. An important practical factor is also the comparison of mechanical properties and the nature of fracture for all the main types of static loading - tension, compression and bending.
  4. The quality of the analysis of the problem and the validity of its relevance in the introductory part of the manuscript are at a good level.
  5. The task has been completely solved.
  6. The results obtained in the article are completely correct and their discussion was carried out in accordance with the existing scientific concepts.
  7. The consistency (consistency) of the presentation does not require revision to the standard requirements of scientific journals.
  8. The literacy of the presentation is good, the style of presentation corresponds to the scientific nature of the material, and the terminology used is accepted.
  9. The design of the article and its volume correspond to the accepted norms, the figures correspond to the stated topic and are appropriate for illustrating the text, most of the literary sources are recent publications in specialized editions. The degree of self-citation is acceptable.

There are no fundamental comments on the content of the article. There are a number of minor remarks on the form of information submission:

  1. In the Introduction section (line 24) it is probably better to use the word “good” instead of the word “preferential”, since there are titanium alloys with even lower modulus of elasticity, approaching the values ​​of the elastic modulus of bone tissue.
  2. In the section "References" it is desirable to provide more complete information on references 35 and 37.

Author Response

Thank you very much for the positive opinion of the reviewer, we will always try to perform the research with the greatest care and we will make every effort to ensure that future publications also receive a positive reception.  

Reviewer 2 Report

Comments on the text entitled

"Assessment of mechanical, chemical, and biological properties of Ti-Nb-Zr alloy for medical applications".

Submitted to Materials (ref Materials 1044318)

 by V. HOPE et al

This text describes the results obtained in a Ti-Zr-Nb alloy, which can be used as biomaterials. Numerous tests were carried out (mechanical, electrochemical, biological) and the results are interpreted using observations made by X-ray diffraction, XPS and scanning electron microscopy.

The text is clear and well written.

The English language is very correct. The references are numerous, recipes and well adapted.

However, I have a number of remarks, questions or comments:

- The authors speak of "annealed, quenched, aged" states. What are the heat treatment conditions that correspond to these states (time, temperature, ...)?

- Are the results of XPS well done on clean surfaces, after removal of contamination products resulting for example from machining operations?

- The strain rates for mechanical tests should be given in s-1, so that they are independent of the geometrical characteristics of the specimens.

- The tensile or compression curves are raw, uncorrected experimental curves. It is essential to present true curves, in order to be able to calculate the exact values of elongations and moduli. Concerning these modulus values, it would be much more accurate to determine them by a complementary technique, for example by US. Indeed, I think that if the authors find quite high Young's modulus values, it is partly due to these experimental errors. Furthermore, are the elongations on the curves really in %?

- X-ray diffraction analyses on samples after deformation would provide very interesting information on the effect of the deformation on the microstructure.

Some remarks, more details:

- Line 103: Ka and not Ka

- Line 327 ???

In conclusion, an interesting work but which needs corrections before a possible publication.

L’email a bien été copié L’email a bien été copié L’email a bien été copié L’email a bien été copié

Author Response

We would like to thank to the Reviewer for his valuable comments. We have taken into account all raised question here follows the detailed answers. Moreover, all changes we have made to the original manuscript.
